# Effects of Physical Properties of Konjac Glucomannan on Appetite Response of Rats

**DOI:** 10.3390/foods12040743

**Published:** 2023-02-08

**Authors:** Chenfeng Xu, Chao Yu, Siqi Yang, Lingli Deng, Chi Zhang, Jiqian Xiang, Longchen Shang

**Affiliations:** 1Enshi Tujia & Miao Autonomous Prefecture Academy of Agricultural Sciences, Enshi 445000, China; 2College of Biological and Food Engineering, Hubei Minzu University, Enshi 445000, China

**Keywords:** konjac glucomannan, physical properties, correlation, satiety, satiation

## Abstract

Dietary fiber has been widely used in designing foods with a high satiating capacity, as the use of satiety-enhancing food is considered to be a promising strategy for combating obesity and the overweight condition. In the present study, partially degraded konjac glucomannan (DKGM) diets with different water-holding capacities, swelling capacities, and viscosities were used to feed rats to investigate the effects of the fiber’s physical properties in regulating the appetite response of the animals. The results showed that the mass and water content of the gastrointestinal chyme increased as the diet’s physical properties were enhanced by the DKGM, which increased the stomach distention of the rats and promoted satiation. Besides, the hydrated DKGM elevated the chyme’s viscosity, and the retention time of the digesta in the small intestine was prolonged significantly, which resulted in an increased concentration of cholecystokinin-8, glucagon-like peptide 1, and peptide tyrosine-tyrosine in the plasma, thus helping to maintain the satiety of rats. Furthermore, the results of the behavioral satiety sequence and meal pattern analysis showed that DKGM in the diets is more likely to reduce the food intake of rats by enhancing satiety rather than satiation, and will finally inhibit excessive weight gain. In conclusion, the physical properties of dietary fiber are highly related to the appetite response, which is a powerful tool in designing food with a high satiating capacity.

## 1. Introduction

Obesity is a high-risk factor for many fatal diseases. Consuming food with a satiety-enhanced capability is expected to contribute to reducing food intake (FI) and losing body weight. Konjac glucomannan (KGM) is a water-soluble dietary fiber isolated from the tuberous roots of the konjac plant (*Amorphophallus konjac K. Koch*), which consists of _D_-glucose and _D_-mannose units through a β-1, 4-linkage in a molar ratio of 1:1.4~1:1.6. The acetyl-substituted residues scattered randomly in the KGM backbone are highly associated with the water solubility of KGM [1]. This fiber performs well in forming gels, absorbing water, thickening, etc., and is widely used as a food additive or ingredient [2,3]. Nowadays, KGM is considered a satiety-enhancing agent and is favored by researchers in designing and developing satiety-enhanced foods [4]. Actually, KGM is usually used for making konjac tofu, konjac vermicelli, konjac noodles, and derivative products in many Asian countries [5,6,7]. These common foods are quite favored by consumers, especially those with body weight management needs, for these foods are lower in energy density, but higher in satiating capacity. Meanwhile, the anti-obesity activity of KGM has been studied widely, and numerous studies have reported its effect on a food’s glycemic index, on reducing cholesterol and body weight loss, and on regulating gut microbiota [8].

The physicochemical properties of the dietary fibers exhibited during gastrointestinal digestion are supposed to be the final factor in determining the fiber’s nutritional effects or physiological activities [9]. Fiber’s physical properties that are highly related to its nutritional effects mainly include its hydration properties, viscosity, bulking capacity, etc. [10]. Robertson [11] first raised the definition of dietary fiber’s hydration properties in 1999, and the swelling capacity (SC), water-holding capacity (WHC), and water absorption were the three main aspects of fiber’s hydration properties. Specifically, the SC refers to the volume occupied by a known weight of fiber under the conditions used, while the WHC refers to the amount of water retained by a known weight of fiber under a specific condition. Numerous studies have investigated the physiological effects derived from the hydrating properties of dietary fiber, including promoting satiety, inducing fermentation, and promoting bowel movements. Tan et al. [12] investigated soluble fibers with a high WHC and SC on rats’ feeding behavior, and they found that these fibers in the diets contributed to regulating the WHC and SC of gastric digesta and reducing food intake. Besides, Hadri et al. [13] also found that the hydration properties of fiber played an essential role in triggering satiation and maintaining satiety, resulting in a change in an animal’s feeding habits. Furthermore, Carvalho et al. [14] evaluated the effects of replacing meat and fat in beef burgers with different levels of hydrated wheat fiber on the technological characteristics, sensory acceptance, and hunger satisfaction. Their results showed that sandwiches comprised of burgers with 2.5 and 5.0 g of fiber/80 g portion caused the same hunger satisfaction for up to 3 h as the control burgers (containing no fiber). Moreover, fiber with a high WHC and SC could regulate the fermentability of diets, contributing to a lower FI and promoting the growth of beneficial microbiota [15].

The research mentioned above confirmed the effects of fiber’s hydration properties on appetite. Previous studies have revealed that fiber with a high WHC and SC could increase stomach distension, which has been suggested to trigger an afferent vagal signal of fullness and hence contribute to satiation during meals and satiety in the post-meal period [16]. Meanwhile, these fibers could also delay gastric emptying as well as prolong the transit time of food in the intestinal tract, as they could impart a high viscosity to the food matrix, which would contribute to maintaining the feeling of satiety between meals [17,18,19,20]. Nowadays, numerous related studies have mainly focused on the satiety-enhancing ability of fiber; however, few researchers have explored the accurate relationship between fiber’s physical properties and the appetite response of animals. Therefore, based on our previous research results [21], the present study investigated the effects of KGM’s physical properties (WHC, SC, and viscosity) on rats’ FI, digesta retention time, and behavioral satiety sequence. Then, we further explored the detailed relationship between fiber’s physical properties and the appetite response, which would help to understand how appetite could be accurately regulated by applying fiber’s physical properties.

## 2. Materials and Methods

### 2.1. Animals and Diets

A total of 48 male Sprague Dawley rats (10 weeks of age, initial body weight of 392.7 ± 7.2 g) were obtained from SPF (Beijing) Biotechnology Co., Ltd. and housed individually in transparent cages that allowed the recording of FI. The rats were housed in the experimental animal center of Huazhong Agriculture University. The specific pathogen-free animal room was kept at a constant temperature of 25 ± 1 °C with a relative humidity of 60%, and it was illuminated with 12 h dark–light cycles (8:00 on, 20:00 off). The rats were randomly assigned to six groups (n = 8 per group), including a control group (C) and five treatment groups (T1, T2, T3, T4, and T5). The rats were acclimatized for one week with a standard diet fed before the experiment, and they were given free access to feed and water. The experimental procedure was confirmed by the ethics committee of Huazhong Agricultural University (HZAURA-2020–0008).

After the acclimation, the standard feed was replaced with the experimental one. The feed ingredients of different groups were the same except for the kinds of dietary fiber. For the control group, there was 10 wt% of cellulose in the feed, while the counterpart in the treatment groups was partially degraded konjac glucomannan (DKGM) of equal weight. Five kinds of DKGM, marked as DKGM1, DKGM2, DKGM3, DKGM4, and DKGM5, were added to the diets of Groups T1, T2, T3, T4, and T5 sequentially to regulate the physical properties of the diets. Therefore, the physical properties of the diets were modified purposely by adding DKGM with an increased WHC, SC, and viscosity. Commercial konjac flour (KGM > 90 wt%) was kindly provided by Hubei Konson Konjac Gum Co., Ltd. (Wuhan, China). The DKGM was prepared by our lab, and the detailed method is reported in our previous research [21,22]. Briefly, KGM with a definite moisture content was obtained by mixing dry KGM with distilled water. KGM powders with different moisture contents were then placed in an autoclave and hygrothermally treated at different temperatures. By regulating the heating temperature, moisture content, and degradation time, DKGM samples with different molecular weights (MWs) were obtained. The ingredients and nutrient composition of the rat’s feed are listed in Table 1, while the main physical properties of the DKGM and diets are listed in Table 2 and Table 3, respectively.

### 2.2. Behavioral Satiety Sequence

The behavioral satiety sequence (BSS) was analyzed according to the method of Halford et al. [24]. After the acclimation, the rats were fasted overnight for 12 h from 20:00 to 8:00 the next day (Day 1) with water available ad libitum, and they were given a pre-weighed amount of food immediately. Then, the rats’ behaviors were recorded for 2 h (8:00–10:00) with a video system, and the feed leftovers were collected for weighing. The food intake was determined to be the difference between the pre-weighed amount of food and the remaining food at the end of the observation. The behaviors were categorized as follows: feeding (ingesting food, chewing, gnawing, or holding food in paws), drinking, activity (exploring movements or rearing), grooming (scratching, licking, or biting any part of its anatomy), and resting (sitting or lying in a resting position, or sleeping). Data were collated into 10 min-period bins for display [25].

### 2.3. Meal Pattern Analysis

The rats’ behaviors were continuously monitored with the camera system until 8:00 on Day 2 for the meal pattern analysis. During this observation period, the remaining food of each rat was weighed at 20:00 on Day 1 and at 8:00 on Day 2. Meal patterns, including the diurnal (from 8:00 to 20:00) and the nocturnal (from 20:00 to 8:00) periods, were recorded and analyzed according to the method described by Tan et al. [12]. The rats were fed their respective diets ad libitum during the meal pattern analysis. The diurnal and nocturnal FI (g), feeding rate (mg/s), meal size (g), meal duration (s), meal number, and inter-meal interval (min) were recorded. Only an FI higher than 0.3 g with a feeding time lasting longer than 13 s was considered a distinct meal, and the interval between two distinct meals needed to be longer than 10 min [26].

### 2.4. Sample Collection

After the ethology observation, the rats were housed for 21 days and given free access to their respective diet. The FI was recorded daily, and the body weight (BW) of rats was measured every 4 d during this experiment period. Once this diet intervention finished, the rats were fasted from 16:00 until 8:00 the next day, followed by a 2 h ad libitum meal with a recorded FI. Then, the rats were fully anesthetized by inhalation of isoflurane, and blood samples were collected from the ophthalmic venous plexus. The blood was collected in 5 mL heparin sodium-coated tubes and centrifuged at 3000× *g* for 15 min to separate the plasma. The plasma was divided into aliquots and stored in liquid nitrogen for the subsequent analysis. Subsequently, the rats were sacrificed to harvest their digestive tract and epididymal fat pad. The complete stomach and small intestine (containing digesta) were weighed to determine the total weight. Then, the gastric and intestinal digesta were collected and stored in liquid nitrogen for the subsequent analysis. The remaining tissue was flushed with saline and blot-dried with chipless filter paper for weighing. The digesta was then determined as the difference between the total and lean tissue weight.

### 2.5. Physicochemical Analysis

The apparent viscosity of the diets and digesta was tested with a stress-/strain-controlled rheometer (DHR-2, TA Instrument, New Castle, DE, USA) at 37 °C. The shear rate was fixed at 50 s^−1^ (which has been reported as a representative shear rate derived from gastrointestinal motility [27]) to record the sample’s η_50_ value by applying the peak hold procedure. The geometry used was a 6 cm parallel aluminum plate, and the gap was set to 500 μm. The digesta’s viscosity could be examined once it recovered to room temperature after being removed from the liquid nitrogen. For the diets, the viscosity in the extract was used to represent its viscosity. Specifically, 2 g of each diet was dissolved with 8 mL of extraction solution (containing 0.9 wt% NaCl and 0.02 wt% NaN_3_) and extracted for 1 h at 40 °C. They were centrifuged (10,000× *g*, 4 °C) for 20 min to obtain the supernatant fraction, the viscosity of which was then measured with the rheometer [28].

The diets and digesta were freeze-dried and ground to a powder before measuring their SC and WHC. The SC was measured as described by Serena et al. [29]. Briefly, 0.3 g of the sample was dissolved with 10 mL of extraction solution (containing 0.9 wt% NaCl and 0.02 wt% NaN_3_) and extracted for 20 h at 39 °C in a shaking water bath at 150 movements/min. The sample’s swelling volume (mL) was recorded after standing for 1 h. The SC (mL/g) was calculated as the sample’s volume divided by its weight (g). After the SC test, samples in the tubes were centrifuged for 20 min at 10,000× *g* and 4 °C. The tubes were semi-inverted for 30 min to drain the water, and the sediment was weighed. The amount of water (g) held by the samples was determined as the difference between the sediment weight and its dry weight. Then, this calculated water weight (g) was divided by the sediment’s dry weight (g), and the final value was considered the WHC of the samples. 

### 2.6. Mean Retention Time Analysis

The rats’ feed and digesta were freeze-dried and ground to powder. Then, they were digested with a microwave dissolver (CEM Corporation, MARS 6, Boston, MA, USA) and their chromium content was measured with an atomic absorption spectrometer (Agilent Corporation, 204DUO, Santa Clara, CA, USA). Then, the Cr_2_O_3_ content could be calculated accordingly. The MRT was calculated using the following formula:MRT = 24 × C × W/I,(1)
where MRT is the digesta’s mean retention time (h); C is the digesta’s Cr_2_O_3_ concentration (mg/g); W is the dry weight of the digesta (g); I is the Cr_2_O_3_ intake of rats in a day; and 24 is the hours [30].

### 2.7. Chemical Analysis

The chemical analysis was performed as previously described [21]. Appetite-related hormones, including insulin, glucagon-like peptide (GLP-1), peptide YY_3–36_ (PYY_3–36_), cholecystokinin-8 (CCK-8), and ghrelin, were assayed using an enzyme-linked immunosorbent kit (Jiangsu Meimian Industrial Co., Ltd, Yancheng, China), while the blood glucose was measured with a biochemical kit (Shanghai Rongsheng Biotech Co., Ltd., Shanghai, China).

### 2.8. Statistical Analyses

The statistical analyses were conducted using OriginPro 2021 (OriginLab Corporation, Ver. 9.8.0.200, Northampton, MA, USA). The data underwent variance homogeneity and normality tests prior to a one-way ANOVA analysis. Tukey’s post hoc test was applied to compare the means, and a difference with statistical significance was accepted at *p* < 0.05.

## 3. Results and Discussion

### 3.1. FI and Body Weight Change

As shown in Table 4, no significant difference in the BW of rats among the different groups was found after the acclimation. However, the daily average BW gain of the rats in Groups C and T1 was beyond 5 g, which was significantly higher than that of the rats in Groups T2, T3, T4, and T5 during the dietary intervention (*p* < 0.05). After a 3-week intervention, the BWs of the treatment groups (except for the T1 group) were significantly lower than that of the control group. The DKGM added into the diets changed their physical properties and thus may have altered the feeding behavior of the rats, resulting in a significant body weight gain eventually.

The BW change of rats during the dietary intervention is shown in Figure 1A. Compared with the control group, a relatively lower BW gain was observed in the treatment groups, especially for Groups T4 and T5, whose BW growth was obviously slower than that of the other groups. From Figure 1B, we noted that the treatment groups’ weekly FIs were significantly lower than that of the control group (*p* < 0.05). The FI difference was more evident in the first week, and the rats consumed more feed in this stage than in the later periods. The different FI eventually led to various BW changes in the rats. As shown in Figure 1C, the BW gain of the control group was as high as 110 g after the dietary intervention, which was significantly higher than that of the T2, T3, T4, and T5 groups, and the BW gains of the T4 and T5 groups were all lower than 70 g.

The epididymis fat pad is relatively easy to be stripped for observation, and it is usually considered a representative adipose tissue that reflects body fat accumulation in male rats [31]. As shown in Figure 1D, the fat accumulation in the epididymis fat pad was obviously inhibited in rats that were fed diets with enhanced physical properties. For example, in the T4 and T5 groups, the rats’ adipose was significantly less than that in the control and T1 groups (*p* < 0.05). Due to the increased hydration capacity and viscosity of the DKGM, the diets’ satiating capacity increased accordingly, resulting in a lowered FI of rats and thus slowing down the accumulation of the rats’ adipose tissue.

### 3.2. Mass and Water Content of the Digesta

A slight, but not significant, increase in the stomach digesta mass was observed as the diets’ physical properties were enhanced by the DKGM; a difference with statistical significance could only be found between the control group and Groups T3, T4, and T5 (Figure 2A). Maybe this was because the DKGM did not significantly change the diets’ satiating capacity during gastric digestion; thus, the rats’ FI was not altered sufficiently within a limited time. We could speculate that the DKGM presented in the diets was less likely to be hydrated fully, as the water content of the stomach digesta was generally lower than 50% (Figure 2B). Even so, the water content of the digesta increased as the hydration capacity of the DKGM increased, and a significant difference was observed between the control group and Groups T3, T4, and T5 (*p* < 0.05).

The intestinal digesta mass and water content of rats in different groups are shown in Figure 2C,D. The digesta in the small intestine was relatively less than that in the stomach, and no significant difference was observed among the control, T2, and T3 groups. However, as the hydration properties of the DKGM increased, the intestinal digesta of the rats in Groups T2, T3, T4, and T5 increased significantly (*p* < 0.05). Besides, the digesta’s water content in the groups also differed significantly, with an apparent increasing trend. The digesta in the small intestine was digested for a longer time and mixed with adequate digestive juice when compared with the gastric digesta. Therefore, the DKGM presented in the intestinal chyme could be hydrated sufficiently, resulting in the difference in the digesta’s mass and water content for rats in the different groups.

### 3.3. Physical Properties and MRT of the Gastrointestinal Digesta

The physical properties of food during digestion may influence the appetite response through differences in mastication, appetite sensation, gastrointestinal peptide release, and FI [32,33,34]. Therefore, the chyme’s WHC, SC, viscosity, and MRT were examined in the present study. Figure 1A shows the apparent viscosity of the stomach at a shear rate of 50 s^−1^. The DKGM samples of different MWs changed the rheology behavior of the digesta. Therefore, the viscosity of the stomach digesta in the treatment groups was significantly higher than that of the control group, and a statistical difference could also be found between the T1 group and the other treatment groups (Figure 3A). Besides, we noted that the stomach digesta was in the form of a viscous paste, showing an extremely weak fluidity. However, a high fluidity in the intestinal digesta was observed as relatively less feed was transported to the small intestine tract, which was then mixed with adequate digestive juice, resulting in a lower viscosity of the intestinal digesta than that of the stomach digesta. It should not be ignored that the viscosity of the intestinal digesta in both the control and T1 groups was extremely low, with values of 7.3 × 10^−4^ Pa·s and 0.046 Pa·s, respectively. The digesta viscosity increased with the MW of the DKGM added to the diets, leading to a statistically significant difference among each group (*p* < 0.05). The interaction between dietary fibers and the mucus layer would result in localized increases in the viscosity adjacent to the brush border of the small intestine, slowing down the nutrient diffusion across it [35]. Therefore, the transport and absorption rate of nutrients in the intestinal tract would be changed heavily by the chyme with a huge viscosity difference, leading to a different satiety sensation in the rats.

As shown in Figure 3B, the difference in the WHC of the stomach digesta between groups was quite narrow, and statistically significant differences were only observed between the control group and Groups T4 and T5. However, we noted that the WHC of the intestine digesta among the groups differed significantly, and their WHC values were generally greater than those of the stomach digesta. Besides, the intestinal digesta’s WHC increased with statistical differences as the DKGM’s WHC increased. This difference may have resulted from the varied physical properties of the DKGM, as DKGM with a higher hydration capacity would be more likely to be trapped in the intestinal tract, thus leading to a significant difference in the digesta’s WHC.

The SC of the digesta in the stomach and small intestine was also investigated, and the results are shown in Figure 3C. The SC of the gastric digesta in the treatment groups was significantly higher than that of the control group, and statically significant differences were also observed among the treatment groups (*p* < 0.05). Interestingly, we noted that the SC of the stomach digesta in the control group was significantly higher than that of the T1 group, revealing that the insoluble cellulose presented in the feed of the control group also possessed a certain degree of SC. The SC of the intestine digesta in the control group was also higher than that in the T1 group, but without a statistical difference, while the SC in the other treatment groups was significantly higher than in the T1 group (*p* < 0.05). Overall, the SC of the digesta in the small intestine was generally higher than that in the stomach, indicating that the dietary fiber would be more likely to absorb water to swell after gastrointestinal digestion.

We further investigated the effect of DKGM samples that varied in their physical properties on the MRT of the digesta, and the results are shown in Figure 3D. The MRT of the digesta substantially increased as the diet’s physical properties were enhanced by the DKGM. For example, the retention time of the gastric chyme was over 3 h, which was significantly longer than that of the control group, whose MRT was only about 2.5 h. Differences with statistical significance were also observed between Group T5 and other groups. During oral mastication and gastric digestion, the viscosity and volume of the gastric chyme increased as the DKGM hydrated gradually, which then delayed gastric emptying, thereby increasing the chyme’s retention time.

Numerous studies have found that chyme retained in the gastrointestinal tract is highly related to gastric emptying and intestinal motility. Wolever et al. [36] found that as the viscosity of the breakfast meal increased by the addition of oat beta-glucan, a delayed gastric emptying rate was observed in healthy humans. Low et al. [37] evaluated the effects of cereal dietary fibers on the retention of digesta along the gastrointestinal tract. Their results showed that the cereal dietary fibers and wheat arabinoxylan delayed the gastric emptying of solid and liquid contents and prolonged their retention time in the small and large intestines. Besides, their results revealed that the apparent MRT was dependent on the structure and swelling capacity of the added cereal dietary fiber.

In addition, the MRT of the intestinal digesta was generally more prolonged than that of the gastric digesta, and the intestinal digesta’s MRT in the treatment group, except for the T1 group, was significantly longer than that of the control group (*p* < 0.05). In the small intestine tract, the DKGM was further hydrated and bonded to a large amount of water, leading to an increased viscosity and bulk volume of the digesta. The transportation of the chyme in the intestinal tract was delayed accordingly, thereby increasing the chyme’s MRT, which would contribute to maintaining the animal’s satiety.

### 3.4. Appetite Biomarkers

The content of the plasma appetite-related biomarkers of rats 2 h after feeding was examined, with the results listed in Table 5. Even though an increasing trend in glycemic concentration was quite evident, the difference between groups was quite narrow, with a significant difference only observed between the control and T5 groups. However, it was found that the concentration of other appetite hormones changed obviously as the chyme’s state in the gastrointestinal tract was regulated by the DKGM. Research has found that the L-cells located in the distal part of the small intestine can respond to stimulation by nutrients (such as carbohydrates and fat) and then release the satiety hormones GLP-1 and PYY_3–36_ [38,39]. Therefore, DKGM with a greater hydration capacity would be more likely to facilitate the retention of chyme in the distal small intestine, resulting in the increased release of GLP-1 and PYY_3–36_. As shown in Table 4, the GLP-1 concentration of the rats in Groups T3, T4, and T5 was significantly higher than that of the control, T1, and T2 groups. In contrast, the differences as well as the increasing trend of PYY_3–36_ concentration in the groups were relatively more evident. Specifically, the PYY_3–36_ concentration of the rats in the treatment groups (except for the T1 group) was significantly higher than that in the control group, and differences with statistical significance also appeared among the treatment groups. Besides, the concentration changes in insulin, ghrelin, and CCK-8 also confirmed the effect of fiber’s physical properties on the appetite response, whereby fiber with enhanced physical properties would increase the satiety and decrease the hunger of the rats.

### 3.5. Behavioral Satiety Sequence

We explored the effects of DKGM with different physical properties on regulating the feeding behavior of rats, and the results are shown in Figure 4 and Figure 5 and Table 5. The overall pattern of behaviors was quite similar among all the groups, which were generally in the form of eating first and then resting. Even though the time that the rats in each group completely stopped eating was all before Time Bin 7, slight differences in the satiety point were also observed among the groups. Specifically, the satiety point of the control group appeared before Time Bin 5, while those of all the treatment groups appeared after the time bin. Especially for the rats in the T5 group, the satiety point was almost in Time Bin 4. These results imply that diets with enhanced physical properties could raise the rats’ satiation, and their intensity was strong enough to inhibit the feeding behavior within a relatively limited time.

The detailed frequency of feeding and non-feeding behavior of the rats in each observing unit (10 min × 12 bins) is shown in Figure 4. The activity and drinking frequency of rats in each time bin was moderately stable, and the grooming and resting behavior generally occurred in the later stages of the observation. Besides, the rats mainly ate before Time Bin 6 and spent most of their time resting after Time Bin 6. For example, the time proportion of eating in Time Bin 2 for the rats in Groups C, T1, T2, T3, T4, and T5 was 84.67%, 79.83%, 78.83%, 81.67%, 80.67%, and 81.67%, respectively, and no statistically significant differences were observed between groups (*p* > 0.05). However, the time proportion in Time Bin 5 was 30.17%, 28.33%, 23.50%, 16.67%, 12.00%, and 5.33%, respectively. In comparison, the proportions in Time Bin 6 lowered to 16.67%, 7.50%, 1.33%, 1.17%, 1.17%, and 1.00%, respectively. Overall, the proportion of eating time in the time bins decreased gradually as the DKGM hydration capacity increased, and the eating time of the rats in the treatment groups was significantly shorter than that of the control group (*p* < 0.05). This implies that the effect of fiber’s physical properties on the rats’ satiation was remarkable as the fiber hydrated during digestion, which then eventually decreased the rats’ eating behavior.

Similar research has been reported by Paderin et al. [40]. The pectin of tansies and apples was used in their study to explore its effect on feeding behavior and food intake in mice. The structure of tansy pectin is more branched than apple pectin, and it was able to form a very viscous solution. They found that the non-feeding behavior of mice was not affected by this dietary fiber. However, the food intake and time spent feeding were significantly lower in mice treated with tansy pectin by 33% and 47%, respectively.

The feeding and non-feeding behavior of rats during the 2 h observation is shown in Table 6. No significant differences in the total eating time of rats were observed except for the difference between the control and T5 groups. However, it still could be found that the eating time tended to decrease as the physical properties of the DKGM were enhanced. At the same time, no significant differences were observed among the groups for non-feeding behavior, such as drinking, activity, and grooming. In addition, the resting time of rats tended to increase as the physical properties of the DKGM became enhanced, especially for Groups T4 and T5, whose resting times were significantly longer than those of the control, T1, and T2 groups. Based on the results in Figure 5 and Table 4, it can be concluded that diets with enhanced physical properties would be more conducive to maintaining the rats’ satiety, resulting in an increased resting time after feeding.

### 3.6. Meal Pattern

The meal pattern of rats is shown in Table 7. The whole-day FI of the rats in the treatment groups was significantly lower than that of the control group (*p* < 0.05). It should be noted that differences with statistical significance were only found for the nocturnal, and not the diurnal, food intake among groups. As shown in the table, the nocturnal FI of rats decreased with enhanced physical properties of the diets, and the rats’ FI in the treatment groups lowered significantly compared to that of the control group. The experiment also found that the rats were relatively active at night, while they spent more time resting during the day. Therefore, the DKGM samples that varied in physical properties were more likely to cause a difference in the nocturnal feeding behavior of rats.

When considering the feeding rate, no significant difference was found, whether day or night. These consistent feeding rates indicated that the dietary fiber added to the feeds did not produce an evident effect on the feed palatability or flavor. In addition, each group’s meal sizes and durations were relatively consistent without significant differences, implying that DKGM samples of different MWs did not affect the rats’ satiation immediately, thereby changing the rats’ meal size or meal duration. This was mainly because the dietary fiber in the stomach chyme did not become hydrated and did not swell fully in a limited time, whereby the stimulation intensity of feeds to the rat stomach wall was quite similar [41]. Therefore, the satiation sensations of the rats in each group were close to each other, resulting in a relatively consistent meal size and meal duration among different groups.

Interestingly, significant differences in the meal number of the rats in different groups were observed, including the total, diurnal, and nocturnal meal numbers. The total meal number of rats in the treatment groups, except for the T1 group, was significantly lower than that of the control. Besides, we noted that the meal number of rats decreased with enhanced physical properties of the DKGM, with the difference in the nocturnal meal number being relatively more evident than the diurnal one. As the DKGM enhanced the diet’s satiating capacity, we noted that the meal interval was prolonged significantly. The meal interval of the rats in the T2, T3, T4, and T5 groups was significantly longer than that of the control and T1 groups (*p* < 0.05). Overall, the relatively consistent meal rate and duration, combined with the decreased meal number and the prolonged meal interval of the rats in each group, eventually decreased the food intake of the rats fed with satiety-enhanced diets.

According to these results, we found that DKGM samples with a greater MW would impart chyme with a higher WHC, SC, and viscosity, thereby enhancing the diet’s satiety and satiation capacity. However, it should be noted that both satiation and satiety are the key factors in appetite control. Satiation, known as intra-meal satiety, is the signals or processes that lead to the termination of eating. In contrast, satiety is known as inter-meal satiety, which is the signals or processes that lead to the inhibition of further eating [42]. The present study revealed that the DKGM in the feed could not become fully hydrated immediately to change the rat’s satiation, and its effects reflected in the meal pattern were the relatively consistent meal size and duration for the rats in the different groups. However, as the diets ingested by the rats were continually mixed with the digestive juice, the DKGM then became hydrated and imparted a high viscosity to the chyme in the small intestine. This effect would contribute to maintaining the rats’ satiety, which was confirmed by the decreased meal number and the prolonged meal interval in the rats’ meal pattern. In conclusion, as the DKGM’s hydration rate was quite limited, the feeding behavior of the rats would be more likely to be inhibited by maintaining satiety rather than enhancing satiation.

### 3.7. Correlation Analysis

A coefficient matrix revealing the correlations between variables is shown in Figure 6. Pearson’s correlation coefficient was used to assess the bivariate association degree. Generally, an |r| (absolute value of r) between 0.9 and 1.0 indicates that the variables can be considered very highly correlated; 0.7 ≤ |r| ≤ 0.9 indicates that the variables can be considered highly correlated; 0.5 ≤ |r| ≤ 7 indicates that the variables can be considered moderately correlated; 0.3 ≤ |r| ≤ 5 indicates variables that have a low correlation; and 0 ≤ |r| ≤ 0.3 indicates that the variables have little, if any, (linear) correlation [43]. However, before interpreting the result, we should acknowledge that the correlation does not explain causation directly. Our conclusion should not totally depend on the Pearson correlations; they are only provided as supportive or hypothesis-generating data.

As shown in Figure 6, we found that the diets’ SC and viscosity seemed to be more correlated with the variables observed in rats when compared with the WHC. When considering the correlations between the appetite hormone and the diet’s physical properties, a moderately negative correlation was observed only between the ghrelin and the physical properties, and the other appetite biomarkers were all positively correlated with the diet’s physical properties. Specifically, it was noted that the SC and viscosity were highly correlated with insulin and PYY_3–36_, while the physical properties were moderately correlated with GLP-1 and CCK-8. However, only a low correlation was found between blood glucose and the physical properties. Besides, even though the diets’ physical properties were found to affect the rats’ FI and BW gain greatly, only a low negative correlation was revealed between them. Especially for the diet’s WHC, it was correlated with the rats’ FI and BW with correlation coefficients of −0.302 and −0.169, respectively. Diets with enhanced physical properties did inhibit excessive body weight gain in the rats; however, the path between a diet’s physical properties and the rats’ body weight gain was quite complex, so it is still difficult to accurately predict the weight gain of animals only based on food properties. Interestingly, the diets’ physical properties, especially the SC and viscosity, were found to be highly positively correlated with the chyme’s MRT. Furthermore, a negatively moderate correlation was shown between the overall meal number and the diets’ physical properties, while the overall meal interval was found to be moderately positively correlated with the diets’ SC and viscosity. This result further confirmed that DKGM samples with enhanced physical properties primarily affected the rats’ satiety rather than satiation, leading to a decreased whole-day meal number and prolonged meal interval, thereby lowering the rats’ food intake eventually.

The effects of fiber’s physical properties on appetite regulation are receiving increased interest [44,45]. However, the exact correlation between physical properties and their nutritive effects still requires continuous input from researchers. Based on the obtained data, we explored the correlations between DKGM’s physical properties and their effects on appetite regulation, which would be significant in guiding the design and development of satiety-/satiation-enhanced food.

## 4. Conclusions

The present study explored the effects of a meal’s physical properties regulated with KGM on the appetite response of rats. KGM with an increased WHC and SC can promote an increased mass and water content of the chyme in the GI tract, which contributes to triggering the satiation sensation. The hydrated KGM would elevate the chyme’s viscosity, which would not only delay gastric emptying and lower the ghrelin level, but also prolong the chyme’s retention time in the small intestine, increasing the secretion of CCK-8, GLP-1, and PYY_3–36_, which would thereby enhance the rat’s satiety. Besides, the results of the meal pattern analysis revealed that KGM samples with enhanced physical properties would be more likely to increase a rat’s satiety rather than its satiation. Even though the diet’s SC and viscosity usually showed a more evident correlation with the rats’ appetite response when compared with the diet’s WHC, it is still difficult to accurately predict the weight gain of animals based only on food properties. On the other hand, developing satiety-enhancing foods has been considered a promising strategy for reducing food intake and promoting weight management, and the effectiveness of fiber’s physical properties on appetite regulation has been continuously confirmed by related studies. In the present study, DKGM samples that varied in their WHC, SC, and viscosity were used to explore the appetite regulation pattern of physical properties, which is expected to provide a theoretical reference in weight management.

## Figures and Tables

**Figure 1 foods-12-00743-f001:**
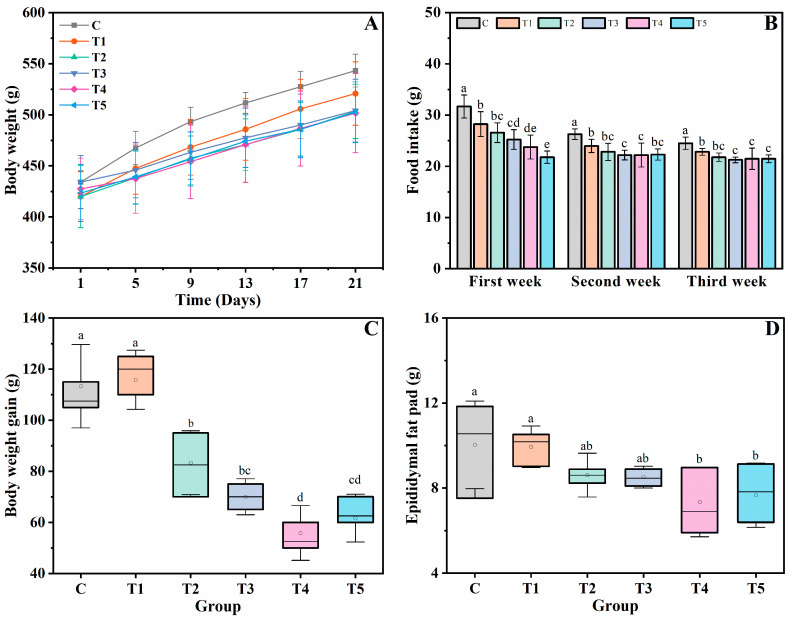
FI and BW of rats. (**A**) Weight growth curve, (**B**) mean weekly FI, (**C**) BW gain, and (**D**) epididymal fat pad mass. Different lowercase letters in the columns reveal a significant difference at *p* < 0.05. Data are shown as means ± SD; n = 8 per group.

**Figure 2 foods-12-00743-f002:**
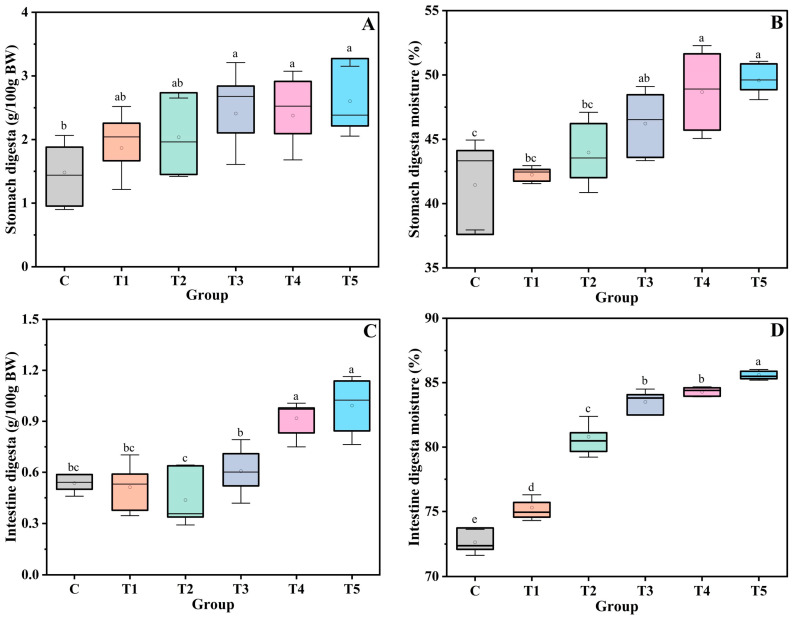
Mass and moisture content of the rats’ digesta. Subfigures (**A**) and (**C**) depicted the stomach and intestine digesta content, respectively, while subfigures (**B**) and (**D**) depicted the moisture of the stomach and intestine digesta, respectively. Means with different lowercase letters differed significantly at *p* < 0.05. Data are depicted as means ± SD; n = 8.

**Figure 3 foods-12-00743-f003:**
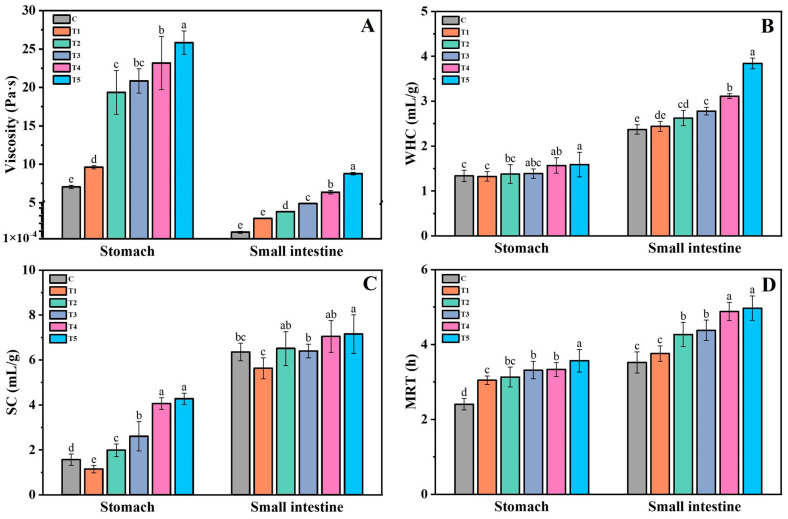
The apparent viscosity (**A**), WHC (**B**), SC (**C**), and MRT (**D**) of the gastrointestinal contents. Means with a different lowercase letter within one group differed significantly at *p* < 0.05. Values are presented as means ± SD (n = 8).

**Figure 4 foods-12-00743-f004:**
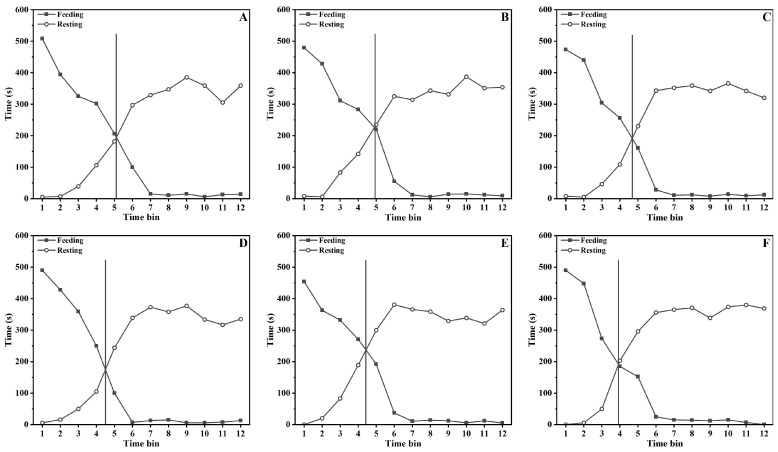
Effect of physical properties of the DKGM on rats’ BSS. Subfigures (**A**–**F**) depicted the BSS of rats in groups C, T1, T2, T3, T4, and T5, respectively. Values are means of n = 8. The perpendicular line of the graph indicates the satiety point.

**Figure 5 foods-12-00743-f005:**
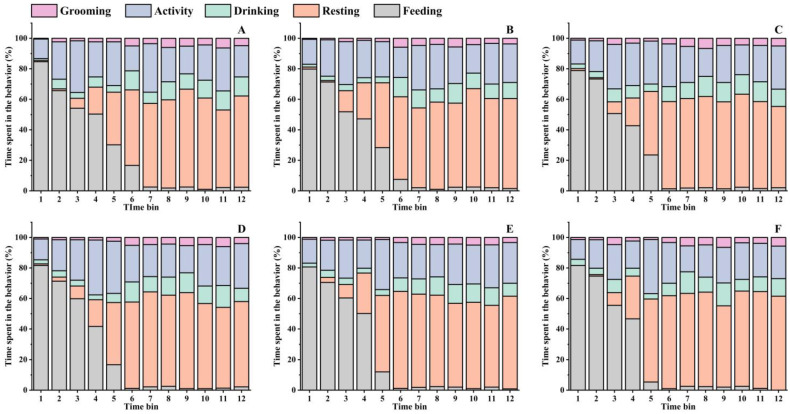
BSS data were collated into 10 min time bins, which are depicted as a percentage of the total behavior. Subfigures (**A**–**F**) depicted the behavior of rats in groups C, T1, T2, T3, T4, and T5, respectively. Columns with different colors represent the time proportion of rats spent in different behaviors.

**Figure 6 foods-12-00743-f006:**
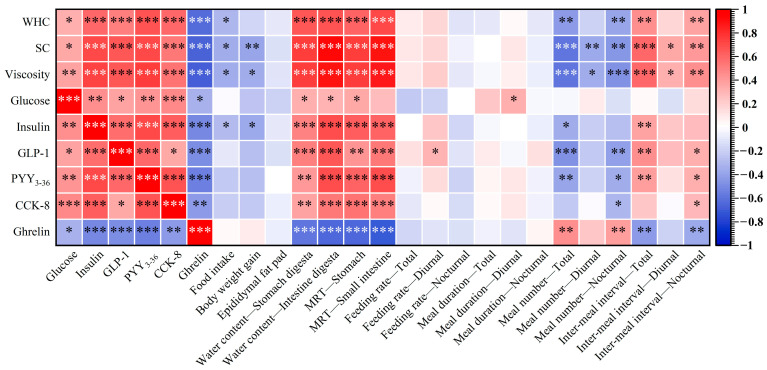
Correlation coefficient matrix revealing the correlations between a diet’s physical properties and the rats’ appetite response. Note: ***, **, and * indicate significance at *p* < 0.001, 0.01, and 0.05, respectively.

**Table 1 foods-12-00743-t001:** Ingredient and nutrient composition of the daily diet for rats ^1^.

Ingredient (g)	C	T1	T2	T3	T4	T5	kCal
Casein	204.8	204.8	204.8	204.8	204.8	204.8	819.2
L-cystine	3.1	3.1	3.1	3.1	3.1	3.1	12.4
Corn starch	331.3	331.3	331.3	331.3	331.3	331.3	1325.2
Maltodextrin 10	135.2	135.2	135.2	135.2	135.2	135.2	540.8
Sucrose	102.4	102.4	102.4	102.4	102.4	102.4	409.6
Cellulose	100	-	-	-	-	-	-
DKGM1	-	100	-	-	-	-	-
DKGM2	-	-	100	-	-	-	-
DKGM3	-	-	-	100	-	-	-
DKGM4	-	-	-	-	100	-	-
DKGM5	-	-	-	-	-	100	-
Soybean oil	71.7	71.7	71.7	71.7	71.7	71.7	645.3
AIN-93 vitamin mix ^1^	10	10	10	10	10	10	40
AIN-93 mineral mix ^2^	35	35	35	35	35	35	0
Choline bitartrate	2.5	2.5	2.5	2.5	2.5	2.5	0
Chromium trioxide	4	4	4	4	4	4	0
Total	1000	1000	1000	1000	1000	1000	3792.5

^1, 2^ Detailed ingredient information about the AIN-93 vitamin mix and AIN-93 mineral mix is shown in the report by Reeves (1997) [23].

**Table 2 foods-12-00743-t002:** Main physical properties of the DKGM used in the rats’ feed ^1^.

Physical Property	DKGM1	DKGM2	DKGM3	DKGM4	DKGM5
Molecular weight (kDa)	157.3 ± 13.46	224.7 ± 8.25	300.7 ± 11.01	455.3 ± 8.20	576.8 ± 16.15
WHC (g/g)	10.29 ± 0.05	10.32 ± 0.12	14.38 ± 0.18	15.08 ± 0.06	15.52 ± 0.13
SC (mL/g)	8.46 ± 0.00	10.39 ± 0.07	15.73 ± 0.10	17.95 ± 0.39	19.67 ± 0.10
η_50_ (Pa·s)	0.06 ± 0.01	0.34 ± 0.06	0.81 ± 0.17	1.64 ± 0.39	2.60 ± 0.66

^1^ Values of DKGM’s physical properties are presented as the means ± SD (n = 3).

**Table 3 foods-12-00743-t003:** Main physical properties of the daily diets for rats ^1^.

Physical Property	C	T1	T2	T3	T4	T5
WHC (g/g)	1.71 ± 0.03	1.69 ± 0.01	1.74 ± 0.01	1.84 ± 0.08	2.04 ± 0.23	2.57 ± 0.16
SC (mL/g)	1.04 ± 0.00	1.66 ± 0.00	3.53 ± 0.01	4.80 ± 0.00	6.80 ± 0.21	6.04 ± 0.02
η_50_ (Pa·s)	0.09 ± 0.00	1.11 ± 0.01	2.50 ± 0.02	6.72 ± 0.09	8.76 ± 0.13	10.63 ± 0.07

^1^ Values of diet’s physical properties are presented as the means ± SD (n = 3).

**Table 4 foods-12-00743-t004:** Effect of the DKGM with different physical properties on rats’ FI and BW ^1^.

Group	Initial BW (g)	Final BW (g)	Daily FI (g)	Daily BW Gain (g)
C	424.17 ± 11.14 ^a^	557.50 ± 25.84 ^a^	27.50 ± 2.55 ^a^	5.21 ± 0.37 ^a^
T1	420.00 ± 14.29 ^a^	535.83 ± 29.05 ^ab^	25.00 ± 2.18 ^b^	5.42 ± 0.34 ^a^
T2	420.01 ± 10.50 ^a^	503.33 ± 26.58 ^bc^	23.80 ± 2.10 ^bc^	4.13 ± 0.80 ^b^
T3	426.17 ± 15.96 ^a^	504.17 ± 30.73 ^bc^	21.98 ± 0.98 ^cd^	3.46 ± 0.37 ^c^
T4	427.50 ± 10.12 ^a^	483.33 ± 37.37 ^c^	22.24 ± 1.07 ^cd^	3.33 ± 0.49 ^c^
T5	423.33 ± 8.05 ^a^	485.00 ± 33.91 ^c^	21.56 ± 0.40 ^d^	3.21 ± 0.43 ^c^

^1^ Values are shown as means ± SD; n = 8 per group. Means with different superscript letters in the same column are significantly different at *p* < 0.05.

**Table 5 foods-12-00743-t005:** Appetite biomarker concentration in rats 2 h after feeding ^1^.

Group	Glucose (mmol/L)	Insulin (mU/L)	GLP-1 (pmol/L)	PYY_3–36_ (pg/mL)	CCK-8 (pmol/L)	Ghrelin (ng/L)
C	9.60 ± 1.40 ^b^	18.82 ± 0.62 ^c^	2.06 ± 0.20 ^c^	16.73 ± 1.03 ^e^	30.15 ± 2.19 ^c^	401.57 ± 17.98 ^a^
T1	9.67 ± 0.98 ^ab^	21.45 ± 2.31 ^b^	2.05 ± 0.14 ^c^	17.95 ± 1.63 ^de^	32.82 ± 1.32 ^b^	362.07 ± 25.99 ^ab^
T2	9.95 ± 0.89 ^ab^	21.78 ± 2.24 ^b^	2.24 ± 0.19 ^bc^	19.05 ± 1.41 ^cd^	34.54 ± 2.78 ^b^	353.65 ± 30.41 ^b^
T3	10.32 ± 0.84 ^ab^	24.04 ± 1.30 ^a^	2.39 ± 0.35 ^ab^	20.24 ± 1.45 ^bc^	33.20 ± 3.56 ^b^	343.65 ± 30.41 ^bc^
T4	10.44 ± 0.81 ^ab^	24.70 ± 3.12 ^a^	2.48 ± 0.26 ^a^	20.99 ± 2.23 ^ab^	34.71 ± 1.79 ^b^	332.58 ± 31.23 ^bc^
T5	10.72 ± 1.34 ^a^	25.66 ± 2.27 ^a^	2.51 ± 0.18 ^a^	22.69 ± 2.24 ^a^	37.35 ± 1.44 ^a^	306.63 ± 21.61 ^c^

^1^ Values are means ± SD (n = 8). Means with a different lowercase letter in the same column indicate a significant difference at *p* < 0.05.

**Table 6 foods-12-00743-t006:** BSS of rats given DKGM with different physical properties ^1^.

Group	Frequency of Feeding and Non-Feeding Behaviors (% Time)				
	Feeding	Resting	Drinking	Activity	Grooming
C	26.19 ± 2.34 ^a^	38.26 ± 3.37 ^c^	8.34 ± 2.79 ^a^	23.84 ± 2.58 ^a^	3.37 ± 2.91 ^a^
T1	24.84 ± 2.63 ^ab^	38.33 ± 2.23 ^c^	8.84 ± 2.77 ^a^	23.45 ± 3.16 ^a^	4.55 ± 2.61 ^a^
T2	24.41 ± 3.40 ^ab^	39.48 ± 2.54 ^c^	9.41 ± 2.78 ^a^	23.09 ± 3.08 ^a^	5.62 ± 3.17 ^a^
T3	23.52 ± 2.43 ^ab^	39.65 ± 2.53 ^bc^	8.31 ± 3.42 ^a^	23.12 ± 2.06 ^a^	5.42 ± 3.54 ^a^
T4	23.21 ± 3.25 ^ab^	41.77 ± 2.39 ^ab^	7.74 ± 2.94 ^a^	22.22 ± 2.75 ^a^	5.04 ± 4.02 ^a^
T5	23.06 ± 3.61 ^b^	42.43 ± 2.40 ^a^	8.46 ± 3.02 ^a^	21.67 ± 4.99 ^a^	4.38 ± 2.37 ^a^

^1^ Values are shown as means ± SD (n = 8). Means with different superscript letters in one column are significantly different at *p* < 0.05.

**Table 7 foods-12-00743-t007:** Meal pattern of rats fed with different DKGM samples ^1^.

Item	C	T1	T2	T3	T4	T5
Food intake (g/d)						
Total	32.2 ± 1.7 ^a^	30.5 ± 2.8 ^ab^	28.9 ± 2.1 ^ab^	27.9 ± 3.0 ^b^	23.9 ± 3.8 ^c^	24.2 ± 3.3 ^c^
Diurnal	13.7 ± 1.4 ^a^	13.9 ± 1.7 ^a^	12.5 ± 1.88 ^a^	13.8 ± 1.5 ^a^	11.8 ± 2.8 ^a^	13.1 ± 1.7 ^a^
Nocturnal	18.4 ± 0.6 ^a^	16.6 ± 1.2 ^b^	16.5 ± 1.0 ^b^	14.1 ± 1.8 ^c^	12.2 ± 1.3 ^d^	11.1 ± 1.2 ^d^
Feeding rate (mg/s)						
Total	4.6 ± 0.7 ^a^	4.0 ± 0.2 ^a^	4.5 ± 0.6 ^a^	4.6 ± 0.5 ^a^	4.4 ± 1.1 ^a^	4.4 ± 0.5 ^a^
Diurnal	4.3 ± 1.7 ^a^	3.9 ± 0.7 ^a^	3.8 ± 0.8 ^a^	4.9 ± 1.8 ^a^	4.4 ± 1.6 ^a^	4.9 ± 1.2 ^a^
Nocturnal	5.7 ± 0.6 ^a^	4.2 ± 0.7 ^a^	5.4 ± 1.3 ^a^	4.9 ± 1.0 ^a^	4.7 ± 1.5 ^a^	4.8 ± 0.8 ^a^
Meal size (g/meal)						
Total	1.5 ± 0.2 ^a^	1.4 ± 0.2 ^a^	1.5 ± 0.2 ^a^	1.6 ± 0.3 ^a^	1.5 ± 0.3 ^a^	1.5 ± 0.3 ^a^
Diurnal	1.3 ± 0.4 ^a^	1.3 ± 0.3 ^a^	1.4 ± 0.3 ^a^	1.6 ± 0.4 ^a^	1.5 ± 0.5 ^a^	1.6 ± 0.4 ^a^
Nocturnal	1.8 ± 0.6 ^a^	1.6 ± 0.4 ^a^	1.8 ± 0.4 ^a^	1.7 ± 0.3 ^a^	1.4 ± 0.4 ^a^	1.5 ± 0.7 ^a^
Meal duration (s)						
Total	327.7 ± 69.7 ^a^	350.2 ± 42.5 ^a^	324.8 ± 28.9 ^a^	346.4 ± 44.1 ^a^	345.9 ± 75.5 ^a^	326.8 ± 39.6 ^a^
Diurnal	320.2 ± 93.8 ^a^	322.9 ± 46.0 ^a^	351.3 ± 94.9 ^a^	335.2 ± 108.3 ^a^	353.0 ± 89.2 ^a^	337.1 ± 111.4 ^a^
Nocturnal	334.2 ± 79.4 ^a^	369.6 ± 91.4 ^a^	341.2 ± 92.7 ^a^	345.6 ± 96.4 ^a^	335.2 ± 115.2 ^a^	327.0 ± 95.5 ^a^
Meal number (meals/d)						
Total	21.5 ± 1.0 ^a^	22.0 ± 1.4 ^a^	18.8 ± 1.7 ^b^	17.5 ± 1.4 ^bc^	17.7 ± 1.4 ^bc^	16.8 ± 1.5 ^c^
Diurnal	11.0 ± 1.8 ^a^	11.2 ± 1.0 ^a^	9.0 ± 0.9 ^b^	8.5 ± 0.8 ^b^	8.2 ± 1.0 ^b^	8.3 ± 0.5 ^b^
Nocturnal	11.3 ± 1.2 ^a^	11.5 ± 1.0 ^a^	9.7 ± 1.2 ^b^	9.2 ± 1.3 ^bc^	8.8 ± 0.8 ^bc^	8.2 ± 0.8 ^c^
Inter-meal interval (min)						
Total	61.4 ± 3.2 ^c^	59.9 ± 4.0 ^c^	71.1 ± 6.1 ^b^	77.0 ± 6.2 ^ab^	76.1 ± 7.2 ^ab^	80.7 ± 7.0 ^a^
Diurnal	61.2 ± 11.0 ^b^	59.4 ± 5.8 ^b^	74.8 ± 8.3 ^a^	79.2 ± 10.7 ^a^	83.9 ± 11.2 ^a^	80.4 ± 5.6 ^a^
Nocturnal	32.2 ± 1.7 ^a^	30.5 ± 2.8 ^ab^	28.9 ± 2.1 ^ab^	27.9 ± 3.0 ^b^	23.9 ± 3.8 ^c^	24.2 ± 3.3 ^c^

^1^ Values are means ± SD; n = 8 per group. Means in each row with a lowercase letter differed significantly at *p* < 0.05.

## Data Availability

Data are not available in public datasets, please contact the authors.

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
