# Peer review of "Effects of Physical Properties of Konjac Glucomannan on Appetite Response of Rats"

_foods, 2023, doi:10.3390/foods12040743_

Round 1

Reviewer 1 Report

The aim of this study was to evaluate the effect of the physical properties that konjac glucomannan added to food has on the appetite response in rats. In my opinion, the methods are clearly and concisely described and a large amount of experimental data was 

obtained, which from my point of view is clearly presented and well discussed.

Some minor issues:

 In my opinion, the title of the manuscript does not reflect its objective well. The study not only assesses the effect that konjac glucomannan added to meals has on the appetite response in rats, but also studies the effect modified konjac glucomannan has on appetite, which should be better reflected in the title.

 Lines 51, 54 and 56: bibliographical references are missing in the text.

Figure legends could be clearer. For example, figures 2B and 2D have both axes equal and the legend does not specify which are the results presented by each one. Also in Figure 3, the legend is not uniform since it specifies which property is associated with figure 3A but not to the rest of the properties. In my opinion it would be clearer if all the legend maintained the same criteria. The same occurs in Figure 5 where the legend may not make it clear to readers what data it presents.

Author Response

Thanks for your useful comments and suggestions on our manuscript. We have revised the manuscript accordingly, and A point-by-point response to the Reviewer can be found in the separate file titled “A point-by-point response to Reviewer.”

Reviewer 2 Report

Summary

In this article the authors assess the potential for partially degraded konjac glucomannan (with differing physical properties – molecular weights) to act on the appetite response in rats.

General Comments

The manuscript demonstrates clearly and effectively the responses of the differences incurred by the additions of differing DKGM. Potential reasoning and hypotheses are provided for all findings throughout the manuscript.

A few suggestions from the reviewer would be:

1. The authors should provide a few more references when analyzing results to help back up and support their assumptions.

2. Add a section or at least a sentence of consideration on the effect of konjac glucomannan to the gut microbiome of the rat and potentially how this might affect the human GM (as the end target of this research area is for possible human applications).

3. Instead of writing ‘DKGM varied in physical properties’ the authors should consider an alternative phrasing. Possibly ‘DKGMs of different molecular weight (MW)’ and then simply 'DKGMs of different MW' thereafter, as the molecular weight is what causes the ‘varied physical properties’.

Title

‘Effects of meal physical properties’ is maybe not the most concise title. Consider rewording.

Tables and Graphs

Tables and graphs have no descriptions of no. of replicates (biological or analytical) attached.

Line 28

Suggest removing ‘kind of’

Line 34

What are Konjac fans?

Line 37-38

Consider revising this sentence, the reviewer would consider it not to be completely accurate. The referenced literature is a review of RCTs – with only one reflecting a potentially, slightly higher, beneficial effect.

Line 43

Consider referencing format.

Line 100

The reviewer is not sure it is correct to say the physical properties are enhanced, this seems a little forward, perhaps modified is a more suitable phrasing.

Line 101-102

The reviewer would like to see some information regarding the specific structure of the DKGM – acetylations? Branching? G:M ratio? Residual monomer content? This information does not seem to be available in either the manuscript or the referenced research.

Table 1

The reviewer considers 10 % weight inclusion a high number. Can the authors please comment on this and potentially where it fits into other similar trials with animal/human and fibers?

Line 193

How is Cr2O3 measured?

Figure 1

Can the authors provide any suggestions as to why the data for T4 and T5 seem to be the opposite, in terms of trend, to the expected?

Line 381-382

Is this referencing time bin 6?

Line 390-391

Is there something missing from this sentence?

Author Response

(The authors gave the same response as above.)
